# Risk factors for postpartum haemorrhage in the Northern Province of Rwanda: A case control study

Oliva Bazirete[1]*, Manassé Nzayirambaho[1], Aline Umubyeyi[1], Innocent Karangwa[2], Marilyn Evans[3]

1 College of Medicine and Health Sciences, University of Rwanda, Kigali, Rwanda, 2 University of Cape Town, Cape Town, Republic of South Africa, 3 Western University, London, Canada

* baziretoliva@gmail.com

**Data Availability Statement:** All relevant data are within the paper and its Supporting information files.

## Abstract

### Background

Postpartum haemorrhage (PPH) remains a major global burden contributing to high maternal mortality and morbidity rates. Assessment of PPH risk factors should be undertaken during antenatal, intrapartum and postpartum periods for timely prevention of maternal morbidity and mortality associated with PPH. The aim of this study is to investigate and model risk factors for primary PPH in Rwanda.

### Methods

We conducted an observational case-control study of 430 (108 cases: 322 controls) pregnant women with gestational age of 32 weeks and above who gave birth in five selected health facilities of Rwanda between January and June 2020. By visual estimation of blood loss, cases of Primary PPH were women who changed the blood-soaked vaginal pads 2 times or more within the first hour after birth, or women requiring a blood transfusion for excessive bleeding after birth. Controls were randomly selected from all deliveries without primary PPH from the same source population. Poisson regression, a generalized linear model with a log link and a Poisson distribution was used to estimate the risk ratio of factors associated with PPH.

### Results

The overall prevalence of primary PPH was 25.2%. Our findings for the following risk factors were: antepartum haemorrhage (RR 3.36, 95% CI 1.80–6.26, P<0.001); multiple pregnancy (RR 1.83; 95% CI 1.11–3.01, P = 0.02) and haemoglobin level <11 gr/dL (RR 1.51, 95% CI 1.00–2.30, P = 0.05). During the intrapartum and immediate postpartum period, the main causes of primary PPH were: uterine atony (RR 6.70, 95% CI 4.78–9.38, P<0.001), retained tissues (RR 4.32, 95% CI 2.87–6.51, P<0.001); and lacerations of genital organs after birth (RR 2.14, 95% CI 1.49–3.09, P<0.001). Coagulopathy was not prevalent in primary PPH.

**Funding:** The authors received no specific funding for this publication.

**Competing interests:** The authors have declared that no competing interests exist.

## Conclusion

Based on our findings, uterine atony remains the foremost cause of primary PPH. As well as other established risk factors for PPH, antepartum haemorrhage and intra uterine fetal death should be included as risk factors in the development and validation of prediction models for PPH. Large scale studies are needed to investigate further potential PPH risk factors.

## Background

Maternal mortality remains unacceptably high worldwide [1]. According to the World Health Organization (WHO), approximately 295 000 women died during pregnancy or after child-birth in 2017. The vast majority of these deaths (94%) occurred in low-resource settings, and most could have been prevented [1]. The maternal mortality ratio (MMR) in Rwanda is reported to be 203/100,000 live births [2]. Reduction of maternal mortality has long been a global health priority, and a target in the United Nations (UN) 2030 agenda for Sustainable Development Goals is to reduce the global MMR to less than 70 per 100,000 live births [3].

Most maternal complications develop during pregnancy and many are preventable or treat-able. Complications, such as maternal obesity, curettage in previous pregnancy, hypertensive diseases, haemoglobin (Hb) level less than or equal to 10 g/dL [4] may exist before pregnancy and may pose problems during pregnancy leading to PPH, especially if not managed as part of the woman's care [1]. The definition of Primary postpartum haemorrhage (PPH) as a major cause of maternal mortality and severe morbidity has been evolving over time to help identify the people most likely to have morbidity and hence adopt timely health interventions. In recent past, the "reVITALize program of the American College of Obstetricians and Gynecologists' (ACOG)", which aims to standardize clinical obstetric terminologies, defined PPH as an increasing blood loss of 1,000 mL or blood loss followed by signs and symptoms of hypovolemia within 24 hours after birth [5]. The World Health Organization, defines PPH as blood loss of 500 ml or more following a normal vaginal delivery (NVD) or 1000 ml or more following a caesarean section within 24 hours after birth [6–8]. The later definition is applied in the present study considering local context of Low and Middle income countries including Rwanda. Women who experience severe acute complications like the case of PPH share many pathological and contextual factors in relation to their condition. Therefore, the definition of PPH in the context of this study also relates to WHO near-miss definition: "a woman who nearly died but survived a complication that occurred during pregnancy, childbirth or within 42 days of termination of pregnancy" [9].

WHO [6] indicates that most maternal deaths resulting from PPH occur within the first 24 hours postpartum and are preventable and manageable if appropriate and effective resources are readily available. The World Health Organization works to contribute to the reduction of maternal mortality by increasing research evidence, providing evidence-based clinical and programmatic guidance [1]. Consequently, PPH has received increasing attention as a quality indicator for obstetric care [10]. Recent studies have shown an increasing trend in PPH, but the causes for this increase are still uncertain [11].

It is known that PPH is the consequence of several different factors that can occur in isolation or combination, such as: uterine atony, retained placental tissue, genital tract trauma and coagulation dysfunction (the '4Ts' mnemonic: tone, tissue, trauma, and thrombin) [6, 12]. Most cases of PPH are caused by uterine atony [12, 13] where the loss of myometrial tone

allows maternal blood flow to the placental bed and the bleeding continues unchecked. Conversely, studies conducted in Nigeria and Ethiopia [14–16] demonstrated that the commonest causes of PPH were genital trauma and retained placenta. Primary PPH may develop in women with no risk factors [17] and only about one-third of PPH cases have identifiable risk factors [8]. A growing body of literature has investigated predictors of PPH in different countries. These predictors include: previous PPH [10, 18–20], mother's age 35 years or above [19–22], hypertensive disorders in pregnancy [7, 10, 22, 23], prolonged labour or complication during labour [7, 20, 24, 25], operative vaginal delivery and instrumental vaginal deliveries [20, 26, 27], multiparity [21, 24, 25], multiple pregnancy [10, 19], Hb less than 10gr/dL on admission to labour, ante-partum anaemia [23, 26–28], fundal height of 38cm or above or large baby [19, 23, 26], placenta praevia [29, 30] and induction of labour [25, 31].

A number of studies have also attempted to examine other predictors of PPH: delivery by Caesarean section [19, 31], gestational age of 40 weeks of amenorrhoea and above [8, 23], curettage in prior pregnancy, nulliparous or receiving pethidine in labour [23], gestational diabetes mellitus, body mass index (BMI) of 25 or above before pregnancy [22] and chorioamnionitis [32]. HIV positive status was found to be associated with PPH in a prospective cohort study conducted by Ononge *et al.*, [19] in Uganda.

Severe morbidities associated with PPH include anaemia and need for blood transfusion, disseminated intravascular coagulation, hysterectomy, and renal or liver failure [33]. Women who develop PPH may also suffer from complications including: hepatic failure, acute respiratory distress syndrome, need for open surgery, need for intensive care, disseminated intravascular coagulation, hysterectomy and cardiac arrest [34, 35]. In moderate complications, PPH can lead to minor anaemia, fatigue, depression and separation anxiety [34, 36]. Interest in PPH has predominantly focused on the evaluation of its risk factors, prevention, and treatment [33]. Other studies have attempted to understand the reasons for substandard care in PPH [33, 37], accurate diagnosis [38–40] and identification of potentially severe cases [20]. However, most of available evidence about contextual factors associated with PPH has been studied in settings outside Rwanda.

A nationwide facility-based retrospective cohort study of a maternal death audit conducted in Rwanda [41]; confirmed that 70% of reported maternal deaths were due to direct causes, of which PPH was the leading one (22.7% of all reported cases). These figures demonstrate that the rate of dying due to PPH in Rwanda, remains high compared to average rates of dying due to PPH in developed countries (8%). Guidelines adopted for the prevention and management of PPH [6] are implemented in Rwanda to improve the prevention and management of obstetric complications including PPH [42]. To date, little is known about risk factors of primary PPH investigated through a case control study in a Rwandan setting. Given that, early identification of women at risk of PPH is known to contribute to its prevention [43], the key input of this work is the solution it provides for a proactive prevention of primary PPH in Rwanda.

## Methods

### Ethical considerations

All procedures performed in this study involving human participants were approved by Institution Review Board at the College of Medicine and Health Sciences, University of Rwanda (ethical approval No 439/CMHS IRB/2019) prior to enrolment of participants. Permissions to conduct the study was also obtained from all the health facilities included in this study. Informed written consent was obtained from all participants.

## Study design

The present study used an observational case control study [44, 45] which is part of a larger exploratory sequential mixed-methods study aiming "to explore the factors associated with PPH prevention" and to develop a "risk assessment tool for the prediction and prevention of PPH" among clients of the Northern Province of Rwanda. This case control study was preceded by a scoping review [4], a qualitative descriptive study [43] and the development of a content validated risk assessment tool for the prediction and prevention of PPH (RATP) [46]. The RATP was used to explore PPH risk factors in the present case control study.

## Population and setting

The target population was women aged 18 years or above, admitted to the postpartum ward after a live birth at ≥32 weeks' gestation at the health facilities of the Northern Province of Rwanda during the period January 1st 2020 to June 30th 2020. The selection criteria of health facilities included their level of performance in maternal and newborn health (5362 women gave birth in selected health facilities during the study period), their location (rural versus urban), and the geographical accessibility of the health facilities to clients. Considering that the northern province is characterised by a hilly terrain, some inhabitants have difficulties to reach the health facilities as indicated by participants in a recent study [43]. The study sites were selected by the principal investigator and validated by the research committee. The Northern Province of Rwanda was purposively chosen for being in a rural area where some health centres are hard to access, and for its low uptake of antenatal and postnatal services among childbearing women [47]. We sent invitations to participate to six health facilities: 5 hospitals and one medicalised health centre. Four district hospitals and one medicalised health centre provided permission to conduct the study. Therefore, five health facilities were included in this study while one district hospital was excluded. The permission to conduct the study in the excluded district hospital was not granted in spite of the follow up made on the request. In addition, the same hospital become later a temporally Covid-19 treatment center during the period of data collection. A medicalised health centre is a new level in the health system of Rwanda deployed with capable staff (medical doctor, nurses and midwives, paramedics, anaesthesia, etc. . .). It is the level in between district hospital and health centre, which is equipped to attend to patients with acute life threatening conditions especially obstetric conditions [48].

## Participants and data collection procedures

From the target population, the source population for the present case control study was selected to identify the outcome of interest as suggested by Song and Chung [45]. Hence, women in the postpartum period were our source population. A PPH case in our study is defined as blood loss of 500 ml or above within the first hour which is visually estimated by health care providers observing women who change the blood-soaked vaginal pads 2 times or more within the first hour after birth [6, 49, 50]. Primary PPH is also counted in case woman requires a blood transfusion for excessive bleeding after birth due to clinical symptoms and signs of anaemia or hemodynamic decompensation after birth [10] or with Hb level less than 11 gr/dL especially if symptomatic according the findings from Rwanda Demographic Health Survey 2019–20 [2]. The number of blood transfusions units administered to the client with PPH was described. Women who received a blood transfusion because of postpartum anaemia, without evidence of excessive blood loss after birth were excluded from this study. Women with secondary PPH which is characterised by vaginal blood loss (or lochia discharges) at least 24 hours after birth or six weeks after delivery [51], were also excluded from this study. The attending health care provider estimated the blood loss visually in all five health

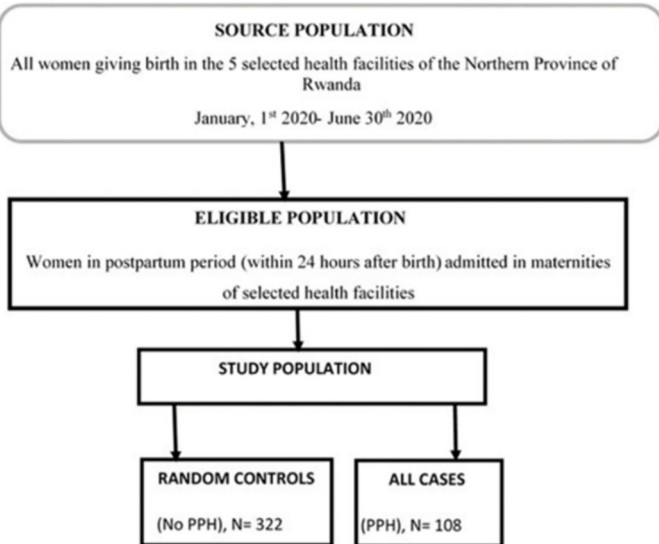

**Fig 1. Selection of study subjects.** PPH: Postpartum haemorrhage.

facilities. Controls were a random sample of all deliveries without primary PPH from the same source population and period of time as the cases of primary PPH. Based on the list in the birth registry, after identifying the case of PPH, the three following women who gave birth without primary PPH were selected as control cases. Hence inclusion criteria were being aged of 18 years and above, being admitted for postpartum monitoring at the health facility within the first 24 hours after birth.

After selecting eligible women for inclusion in the present study, we extracted the study participants [45], that included all cases confirmed to have primary PPH (n = 108) and a random sample of controls without primary PPH (n = 322) (Fig 1) to make a total sample size of 430 participants. Due to a limited study period, the number of study participants was slightly lower than the target sample size. A G*Power software for power analysis [52] indicated that we needed 118 cases and 354 controls considering three controls per case, with type I error of 5%, power of 80%, frequency of risk factors in control subjects of 0.2%, and cases with potential risk factors to PPH being almost twice as likely to be exposed to PPH compared to controls (odds ratio (OR) = 1.8).

The research assistants (one at each of the five health facilities) were registered midwives with experience of working full time in maternity, and were recruited by the principal investigator in agreement with the health facilities' administration. To start data collection, research assistants verbally invited women in postpartum period who met inclusion criteria to take part in the study, told them about the study. Those who agreed to participate signed to indicate informed consent. Data was collected during the hospital stay of the woman to allow the research assistant to visit the client and cross check data. Accessing clients' charts facilitated the assessment of eligibility of study participants. Therefore, registration of client data was based on information collected from clients' files, from maternity records completed on a regular basis on women in labour and birth, also documenting the birth outcomes, including cases that experienced blood loss during birth and immediately after birth. We also collected client data through structured interviews carried out by the research assistant with the women using the RATP to ensure the accuracy of data and to minimise missing data. After identifying a case of PPH, the research assistant was required to also identify three control participants

who gave birth and who did not experience PPH within +/- 24 hours in relation to the time the PPH happened.

The RATP was translated from English to French and Kinyarwanda by a professional translator to facilitate respondents' understanding of the tool by using their preferred language. The three languages are officially used in Rwanda. Back translation was done by an independent professional translator, to confirm that the meaning and content of the questions of the original copy had not been changed during the translation. Verification of the translated instrument was also done to ensure its validity.

As the research assistants were full-time staff working in maternity, they were able to identify cases who experienced PPH after birth during their working days. For some cases of PPH that happened during their days off, the research assistants could identify potential participants through maternity records (daily reports) and confirm whether the woman had primary PPH by asking the woman if she bled heavily and changed the blood-soaked vaginal pads two or more times during the first hour after birth. The principal investigator made regular visits to the research sites during the data collection period to ensure that data collection was being conducted as planned.

## Variables under study

The dependent variable in this study was primary PPH (presence or absence of primary PPH: Case and Controls) while the presence or absence of the potential PPH risk factors among PPH and control cases were the independent variables. The RATP consists of three sections. The first, Section A consist of social and demographic characteristics of the woman: age, marital status, level of education, area of residence, accessibility to nearest health facility, use of medical insurance, use of family planning methods outside pregnancy, health facility where delivery took place, socio economic status and religion. Section B included newborn and mother anthropometry and Hb measurements: newborn weight, woman weight, woman height and woman Hb level. Section C focused on pregnancy, obstetric, intrapartum and immediate postpartum factors: primiparity, multiparity, uterine anomaly, uterine surgery (e.g. myomectomy), previous caesarean section, previous PPH, antepartum haemorrhage, HIV positive status, multiple pregnancy, anaemia, gestational diabetes mellitus, polyhydramnios, anticoagulant medications in pregnancy, severe pre-eclampsia, intra-uterine foetal death, premature rupture of membranes, prolonged labour, spontaneous vaginal delivery, instrumental vaginal delivery, in labour caesarean section, repeat caesarean delivery, labour induction, labour augmentation, administration of oxytocin for active management of the third stage of labour, episiotomy, perineal tear, vaginal wall tear, cervical tear, uterine rupture, retained tissues, uterine atony with full bladder and uterine atony with uterine inversion. Multiparity indicates the clinical case of the woman who has already given birth to more than two babies while grand multiparity is from five babies at a gestational age of 24 weeks or more as defined in literature [53, 54]. For the present study, those with gestational age of 32 weeks and above are included.

## Data analysis

All 430 completed risk assessment tools (108 cases and 322 controls) were captured in an Excel spreadsheet which was exported to STATA version 15.1 to perform data analysis [55]. Data were cleaned to ensure that there were neither errors nor missing data. For data analysis, we distinguished between causes of and risk factors for PPH. Causes of PPH were classified as the '4Ts' mnemonic [12]: Tone (uterine atony, uterine inversion, and full bladder after birth causing PPH), Tissue (retained placenta and retained parts of placenta, and abnormal

placentation), Trauma (uterine rupture, perineal tears and episiotomy, vaginal wall tears, cervical tears), and Thrombin (coagulation disorders, consumption of anti-coagulant medications).

Among the PPH risk factors, maternal age, BMI, birth weight and maternal Hb were recorded as continuous variables for descriptive purposes and for inclusion in the final model for analysis. Maternal age was divided into 4 groups, below 25 (reference group); 25–29; 30–34; 35 and above [56]. BMI was divided into 4 groups as per WHO's recommendation: <18.5; 18.5–24.9 (reference group); 25–29.9; ≥30 kg/m2 [57]. Infant birth weight was grouped into three categories: < 2500g; 2500-4000g (reference group); ≥4000g [58]. Hb level of the client at the time of admission to labour was dichotomized as either anaemic (Hb< 11gr/dL) or non-anaemic (Hb ≥ 11gr/dL) [2].

Data were analyzed using univariate, bivariate and multivariate techniques [59]. Univariate analysis was used first to summarize data in terms of frequency distributions of the variables under study then bivariate was used to examine the relationship between primary PPH (binary outcome variable) and each risk factor/ cause. The relationship was established between outcome variable (developing or not developing primary PPH among childbearing women in selected health facilities) and independent variables (socio-demographic variables and other potential PPH risk factors under study).

Multivariate analysis was conducted to determine to what extent the significant independent variables are in correlation with the outcome variable. The modified Poisson regression model with robust error variances [60] was used to estimate risk ratios (RRs) and 95% confidence intervals (CIs). This model was chosen because the outcome of interest (primary PPH) was common [61, 62]. The absence of PPH was used as the reference category because we hypothesized based on previous research [4, 43] that the likelihood of PPH would be high with the presence of PPH risk factors relative to none. Statistical significance was therefore defined at 95% confidence interval and P-value of <0.05.

Extensive discussion in the literature has reached a consensus that RR is preferred over the odds ratio for most prospective investigations for its scientific meaning [61, 63]. Moreover, odds ratios are considered as more extreme than relative risks when the outcome is not rare [64, 65] (prevalence above 10% in the study population [62, 66]), and conversion of odds ratios into relative risks is known to produce biased estimates when adjusting for covariates [60, 63]. Therefore, Poisson regression, a generalized linear model with a log link and a Poisson distribution was used in this study to estimate the risk ratio because the prevalence of the outcome is not rare (prevalence of primary PPH = 25%) and the outcome variable itself is binary. When the outcome is binary, the exponentiated coefficients are risk ratios instead of incidence-rate ratios [60, 67, 68]. The results are reported in the results section.

## Results

Table 1 shows that PPH cases represent 25.1% of all participants while 74.9% are cases for control. PPH was found in anaemic women. A percentage of 5.6% of PPH cases had received blood transfusion because their intrapartum haemoglobin was less than 7gr/dL while 25.9% were anaemic with haemoglobin between 7-11gr/dL. Of the 6 women who received blood transfusion, 4 received 3 whole blood units of 350ml/client and two received 2 whole blood units of 350 ml/ client. It took more than one-hour walking time for more than half of participants (56.4% for PPH cases, and 50.6% for controls) to reach the nearest health facility and less than one-hour for the rest of participants (43.5% for PPH cases and 49.3% for control group). A majority of participants completed only primary education (83.3% for PPH cases and 79.1% for control group), 11.1% of PPH cases attended secondary education versus 15.5% in control

**Table 1. Characteristics of childbearing women included in this study.**

| Variables | PPH Cases | Controls |
|---|---|---|
| | n = 108(25.1%) | n = 322(74.9%) |
| **Average age**(Ave*) | 32(Ave)* | 29 (Ave)* |
| **Age category** | | |
| <25 | 19(18.3%) | 81(26%) |
| 25–29 | 21(20.1%) | 75(24.1%) |
| 30–34 | 18(17.4%) | 73(23.5%) |
| ≥ 35 | 46(44.2%) | 82(26.4%) |
| **Accessibility to nearest health facility** | | |
| Walking time <1hour | 47(43.6%) | 158(49.4%) |
| Walking time > 1hour | 61(56.4%) | 162(50.6%) |
| **Level of education** | | |
| Never went to school | 6(5.5%) | 17(5.3%) |
| Primary school | 90(83.4%) | 255(79.2%) |
| Secondary school and above | 12(11.1%) | 50(15.5%) |
| **Place of delivery** | | |
| Health Centre | 46(42.5%) | 57(17.7%) |
| District Hospital | 62(57.5%) | 265(82.3%) |
| **Medical insurance** | | |
| No | 18(16.6%) | 10(3.2%) |
| Yes | 90(83.4%) | 312(96.8%) |
| **Multiparity** | | |
| No | 16(14.9%) | 84(26.1%) |
| Yes | 92(85.1%) | 238(73.9%) |
| **AMTSL** | | |
| No | 17(15.9%) | 32(9.9%) |
| Yes | 90(84.1%) | 290(90.1%) |
| **Intrapartum Haemoglobin** | | |
| <7 gr/dL (Anaemic /received blood transfusion) | 6(5.6%) | 0(0%) |
| 7 -11gr /dL (Anaemic/ No blood transfusion) | 28(25.9%) | 28(8.7%) |
| >11 gr/dL (Non Anaemic) | 74(68.5%) | 294(91.3%) |

Ave*: Average.

group. Most of the participants (57.4% for PPH cases versus 82.3% for control group) gave birth at a hospital while others were from the medicalized health centre (42.5% for PPH cases and 17.7% for control group). A large proportion of participants (83.3% of PPH case versus 96.8% of control group) had a medical insurance. A big number of participants (76.7%) were giving birth to a second or subsequent child and 23.2% were primiparous. Most (84.1% of PPH cases versus 90% of control group) received intra-muscular oxytocin after both vaginal and caesarean births to manage the third stage of labour. Univariate analysis demonstrated that the mean maternal age was higher among PPH cases: 32 years (95% CI: 30.9–33.9) vs. 29 years (95% CI: 29.0–30.5) for the control group.

Results from bivariate analysis of selected demographic and clinical characteristics of women from the two groups are shown in Table 2. As indicated, maternal age, possessing medical insurance, health facility where delivery took place, previous PPH, antepartum haemorrhage (APH), multiple pregnancy, anaemia in pregnancy, level of Hb on admission to labour, intrauterine foetal death, and BMI were the significant antepartum risk factors

**Table 2. The number (n), prevalence (%); 95% (CI) and *P*-value of childbearing women with primary postpartum haemorrhage by demographic and clinical characteristics.**

| Variables | Cases(PPH = Yes) | | Controls (PPH = No) | | P-Value |
|---|---|---|---|---|---|
| | **n** | **(%)** | **n** | **(%)** | |
| **PPH risk factors** | | | | | |
| **Age category** | | | | | |
| <25 (Reference) | 19 | 19.0 | 81 | 81.0 | 0.01** |
| 25–29 | 21 | 21.9 | 75 | 78.1 | |
| 30–34 | 18 | 19.8 | 73 | 80.2 | |
| ≥ 35 | 46 | 36.0 | 82 | 64.0 | |
| **Health facility where delivery took place** | | | | | |
| District Hospital | 62 | 18.9 | 265 | 81.1 | |
| Health Center (Reference) | 46 | 44.6 | 57 | 55.4 | <0.001*** |
| **Hemoglobin on admission** | | | | | |
| <11 gr/dl (Anaemic) (Reference) | 34 | 54.8 | 28 | 45.2 | <0.001*** |
| ≥ 11 gr/dl (Non anaemic) | 73 | 20.2 | 288 | 79.8 | |
| **Body Mass Index** | | | | | |
| <18.5 (Reference) | 4 | 80.0 | 1 | 20.0 | 0.01** |
| 18.5–24.9 | 76 | 27.7 | 199 | 72.3 | |
| 25–29.9 | 23 | 17.5 | 109 | 82.5 | |
| ≥30 | 5 | 27.8 | 13 | 72.2 | |
| **Intrauterine foeto-dealth** | | | | | |
| Yes | 11 | 68.7 | 5 | 31.3 | <0.001*** |
| No (Reference) | 97 | 23.3 | 315 | 76.7 | |
| **Medical insurance** | | | | | |
| Yes | 90 | 22.4 | 312 | 77.6 | <0.001*** |
| No (Reference) | 18 | 64.2 | 10 | 35.8 | |
| **Multiparity** | | | | | |
| Yes | 92 | 27.8 | 238 | 72.2 | 0.02** |
| No(Reference) | 16 | 16.0 | 84 | 84.0 | |
| **Previous PPH** | | | | | |
| Yes | 9 | 60.0 | 6 | 40.0 | 0.01** |
| No(Reference) | 99 | 23.9 | 316 | 76.1 | |
| **Antepartum haemorrhage** | | | | | |
| Yes | 10 | 66.6 | 5 | 33.3 | <0.001*** |
| No(Reference) | 98 | 23.7 | 315 | 76.2 | |
| **Multiple pregnancy** | | | | | |
| Yes | 15 | 68.1 | 7 | 31.8 | <0.001*** |
| No(Reference) | 93 | 22.7 | 315 | 77.2 | |
| **Anemia in pregnancy** | | | | | |
| Yes | 11 | 84.6 | 2 | 15.3 | <0.001*** |
| No(Reference) | 95 | 22.8 | 320 | 77.1 | |
| **Premature Rupture of Membranes** | | | | | |
| Yes | 9 | 15.7 | 48 | 84.2 | 0.08* |
| No(Reference) | 98 | 26.5 | 271 | 73.4 | |
| **AMTSL with Oxytocin** | | | | | |
| Yes | 90 | 23.6 | 290 | 76.3 | 0.09* |
| No(Reference) | 17 | 34.6 | 32 | 65.3 | |
| **Causes of PPH** | | | | | |

(*Continued*)

**Table 2.** (Continued)

| Variables | Cases(PPH = Yes) | | Controls (PPH = No) | | P-Value |
|---|---|---|---|---|---|
| | n | (%) | n | (%) | |
| **Uterine atony** | | | | | |
| Yes | 39 | 97.5 | 1 | 2.5 | <0.001*** |
| No(Reference) | 69 | 17.6 | 321 | 82.3 | |
| **Trauma of genital organs** | | | | | |
| Yes | 45 | 30.2 | 104 | 69.8 | 0.08* |
| No(Reference) | 63 | 22.4 | 218 | 77.5 | |
| **Retained tissues** | | | | | |
| Yes | 29 | 96.6 | 1 | 3.3 | <0.001*** |
| No(Reference) | 78 | 19.5 | 321 | 80.4 | |
| **Coagulopathy** | | | | | |
| Yes | 1 | 100.0 | 0 | 0.0 | 0.08* |
| No(Reference) | 107 | 24.9 | 322 | 75.1 | |

Level of Significance at 5% (*slightly significant; **very significant; *** highly significant).

associated with PPH (*P*-value < 0.05). Four risk factors were weakly significant: Multiparity (*P*-value of 0.02), premature rupture of membranes (PROM *P*- value 0.09) and Active Management of Third Stage of Labour (AMSTL) using Oxytocin (p-value 0.09). Differences in the following factors among the two groups (with PPH and without PPH) did not reach statistical significance with a *P*-value greater than 0.05: caesarean birth and spontaneous vaginal birth, positive HIV status, labour induction and labour augmentation, management of the third stage of labour, pre-eclampsia, polyhydramnios and previous uterine surgery. Hence these factors were not correlated with primary PPH and are not shown in Table 2.

At this level of analysis, the causes of PPH that were found to be associated with primary PPH with a *P*-value of <0.001 were uterine atony, and retained tissues. Coagulopathy and lacerations of genital organs were also significantly correlated with PPH but their *P*-value of 0.08 did not meet the 0.05 cut-off set for statistical significance.

The risk factors found to be associated with primary PPH at bivariate analysis were further analyzed to control for possible confounders. This multivariate analysis indicated that the antepartum risk factors were: Hb level of <11 gr/dL on admission to labour, multiple pregnancy, intrauterine foetal death, antepartum haemorrhage and PROM (Table 3). The risk of

**Table 3. Risk ratios, 95% Confidence Interval (CI) of childbearing women with primary postpartum haemorrhage.**

| PPH risk factors/ Causes | Risk ratio | 95% Confidence Interval | P-Value |
|---|---|---|---|
| Haemoglobin <11 gr/dL | 1.519 | [1.000–2.309] | 0.05** |
| Multiple pregnancy | 1.838 | [1.119–3.017] | 0.02** |
| Intrauterine fetal death | 1.937 | [0.931–4.030] | 0.08* |
| Antepartum haemorrhage | 3.362 | [1.805–6.261] | <0.001*** |
| Premature Rupture of Membranes | 0.585 | [0.323–1.058] | 0.08* |
| Uterine atony | 6.701 | [4.784–9.384] | <0.001 *** |
| Retained tissues | 4.326 | [2.871–6.518] | <0.001 *** |
| Genital trauma | 2.149 | [1.491–3.097] | <0.001 *** |

Level of Significance at 5% (*slightly significant; **very significant; *** highly significant).

developing primary PPH is almost two times higher among childbearing women with intra-uterine foetal death (RR 1.9, 95% CI 0.93–4.03, *P* = 0.08); with multiple pregnancy (RR 1.8, 95% CI 1.11–3.01, *P* = 0.02); with haemoglobin <11 gr/dL on admission to labour (anaemic women) (95% CI 1.00–2.30, *P* = 0.05); the risk is three times more in women with APH (95% CI 1.80–6.26, *P*<0.001) than in women without these risk factors. Primary PPH was slightly more prevalent in women experiencing PROM than among those without this condition. (RR 1.5, 95% CI 0.32–1.05, *P* = 0.08) than in women without.

During the intrapartum and immediate postpartum periods, the risk of developing primary PPH is almost seven times greater in childbearing women who experience uterine atony than in births not complicated with uterine atony (RR 6.7, 95% CI 4.78–9.38, *P*<0.001). The risk of PPH was four times higher in women with retained tissues than in those without (RR 4.3, 95% CI 2.87–6.51, *P*<0.001). The risk is twice more in women with genital organ lacerations than in women without this complication (RR 2.1, 95% CI 1.49–3.097, *P*<0.001). The problem of coagulopathy was not significantly correlated with primary PPH.

## Discussion

PPH often occurs in the absence of known risk factors. In the present study, we investigated and modelled the potential risk factors of primary PPH among women admitted to postpartum units of five selected health facilities of the Northern Province of Rwanda.

The prevalence of PPH in our study participants was 25.1%, which indicates that PPH was common in our study population as confirmed by other studies [61, 62]. This result was likely due to the fact that majority of women included in this study gave birth in district hospital settings (76.2%) that receive referrals from health centres of the catchment area. The prevalence of PPH is also noted to be high in Yemen (29.1%) [69]. This variation in prevalence of PPH might be due to difference in study design, social stability, cultural difference and maternal health care services accessibility.

As highlighted by Main *et al.*, [70], the risk assessment for PPH should be undertaken during antepartum care, at admission to labour and delivery, during labour and delivery and post-partum, as PPH risk factors can change or evolve throughout the perinatal period. Our assessment demonstrated that factors associated with primary PPH are found throughout the course of childbearing period; the antepartum, intrapartum and early postpartum periods including causes of PPH for the continuity of care in early detection and prevention of PPH.

### Antepartum risk factors

Antepartum haemorrhage, multiple pregnancy, intrauterine foetal death, Hb level on admission to labour and PROM demonstrated an increased RR for developing PPH which was higher in women with these risks factors than in women without.

Antepartum haemorrhage was the strongest predictor among study participants and was associated with triple the risk for developing primary PPH. This finding concurs with the guideline released by the Royal College of Obsetricians and Gynecologists [71] in the UK, which states that PPH should be anticipated in women who have experienced antepartum haemorrhage [71]. A study in Brazil demonstrated that severe maternal outcome due to ante-partum and intrapartum haemorrhage was highly prevalent [72]. Antepartum haemorrhage might be associated with placenta praevia, placental anomalies and local genital tract disorders, such as cervicitis and neoplasms. No definitive cause is diagnosed in some patients [72]. In our study, insufficient data were available to classify antepartum haemorrhage, and participants were not sure about the kind of antepartum haemorrhage they experienced. A comprehensive

documentation of client obstetrical history and antepartum complications could give light to birth preparedness and early detection of PPH.

Multiple pregnancy was one of the strongest predictors of primary PPH in this study, which is similar to previous studies [10, 19, 31, 73]. The over distension induced by multiple pregnancy increases the risk of uterine atony with overstretching of uterine muscle. The WHO encourages further research to determine the role of symphysis-fundal height measurement in detecting abnormal foetal growth and other risk factors for perinatal morbidity (multiple pregnancy, polyhydramnios, macrosomia) in settings where antenatal ultrasound is not available [74]. In addition, the large placental size in multiple pregnancy increases the insertion surface area which bleeds after childbirth. Multiple pregnancy contributes to an increased fundal height which is shown to be a predictor for PPH [22]. This finding calls for more vigilance on the part of practitioners attending labour and births to identify women at risk, to have adequate preparation and plan for early intervention to prevent PPH. In this regards, as suggested by Fawcus [75], practical approaches to preventing and managing PPH in limited resource settings may be encouraged in the present study like establishment of women groups coordinated by community health worker to enhance birth preparedness and encourage peer support. "Maternity waiting areas" could be also useful in hard to reach areas whereby pregnant women are encouraged to be admitted in lodging facilities close to health facilites waiting for birth and hence allow timely management of possible complications [75].

Intrauterine foetal death (IUFD)—also called stillbirth—was prevalent in participants of our study who experienced primary PPH, again supporting the existing evidence. PPH was observed in 10% of IUFD cases in a US study [76] and in 12% of patients in a study on aetiology and maternal complications of IUFD [77]. In case of stillbirth, PPH might be associated with retained placenta which was noted in a very high number of women (23%) in a retrospective chart review to evaluate stillbirth demographics, pregnancy and maternal risk factors, and complications of labour and birth [76]. Another attributable cause for this association was disseminated intravascular coagulation as reported by evidence [77–79] which definitely lead to massive blood loss. Though global data on causes of stillbirth is limited [80], evidence demonstrates that intrapartum complications, hypertension, diabetes, infection, placental impairment, and pregnancy lasting longer than forty weeks are potential causes of stillbirth [81]; and these causes have been identified to be risk factors for PPH [22, 82]. During antepartum care, it is necessary to track patients with conditions that raise their risk of stillbirth as well as postpartum haemorrhage [83].

Our study was conducted during the Covid-19 outbreak (study period: January–June 2020). Clinicians and caregivers need to be extra vigilant for maternal complications in pregnant women with Covid-19 [76]. Creating an individual care plan for high-risk pregnancies instead of a virtual approach may improve outcomes in this type of situation [78]. Since pregnant women are potentially at risk of obstetric complications including PPH, regular consultation with a health professional is recommended throughout the course of pregnancy, because it is an optimal opportunity for healthcare providers to identify women who are at increased risk early on during pregnancy and to deliver necessary support and educate pregnant women on unexpected events [84]. Rwanda has started implementing the newest guidelines from the World Health Organization recommending the 8 contacts during antenatal period [85]. This is commendable as it is a great opportunity to reduce perinatal morbidity and mortality which includes detection of PPH risk factors and hence improve women's experience of care.

In addition to prevalent risk factors, our bivariate analysis demonstrated significant association of other risk factors with primary PPH with a $P$-value < 0.05. These significant factors included maternal age (being >35 was associated with higher risk), type of health facility

where birth take place, BMI, holding medical insurance, multiparity, previous PPH, anaemia during pregnancy and active management of third stage of labour. The significant antepartum risk factors for primary PPH observed in our study echoed previous evidence [19, 23]. In contrast to previous studies [10, 19, 20, 23], where these same factors were found to be predictors of PPH, our multivariate analysis did not find these variables to be associated with increased risk of primary PPH. This dissimilarity might be associated with difference in sample size and in data collection period. A large scale case control study would investigate further all potential risk factors for improved PPH prevention.

## Intrapartum risk factors and causes of PPH

Low Hb level (<11 gr/dL) during intrapartum period is worth noting because it is an indication of anaemic pregnant women so that health care providers can be proactive in prevention of PPH. For participants included in this study the risk of PPH was 1.5 times higher for anaemic women than for non-anaemic women. Again, our findings are consistent with earlier research [28]. Anaemia in pregnancy is a significant public health problem around the world, particularly in LMIC's, where it is a leading cause of maternal morbidity and mortality [86]. Anemia has long been thought to increase the risk of postpartum hemorrhage [20] and the two conditions together contribute to 40–43% of maternal deaths in Africa and Asia [87]. Severe anemia has been shown to affect myometrial contractility related to diminished transport of hemoglobin and oxygen to the uterus, resulting in tissue enzymes and cellular dysfunction [88]. According to the findings of an observational study demonstrating an association between haemoglobin level at labor and PPH [69], women with Hb of 7gr/dL or less have such a higher risk of PPH due to uterine atony than women with Hb 7.1–10gr/dL. This concur with our findings whereby 5.6% of PPH cases had received blood transfusion because their intrapartum haemoglobin was less than 7gr/dL. Any blood loss after birth that has the potential to cause hemodynamic compromise should be deemed PPH for clinical purposes; and this is likely to happen in conditions like anemia [89].

There are a few studies that link the risk of PPH to anemia levels [69]. Further researches are encouraged to explore more this area. Anemia is prevalent in our country, as it is in other LMICs, especially in hard to reach communities where access to antenatal care services might be difficult. The key findings from Rwanda Demographic Health Survey 2019–20 demonstrated that among pregnant women, those in the lowest wealth quintile are more likely to be anaemic (Hb<11 gr/dL) than other women [2]. Our multivariate analysis also demonstrated PROM to be prevalent in women who developed primary PPH which was identified as potential risk factor in earlier case control study [10]. PROM is complicated with chorioamnionitis which was found to be associated with an increased risk of severe atonic PPH in previous studies [32, 90].

For other intrapartum and immediate postpartum factors analyzed in this study, with the exception of coagulopathy, the other causes of PPH demonstrated a strong association with primary PPH (uterine atony, trauma of genital organs and retained tissues). In this study, uterine atony was found to be the most prevalent cause of primary PPH, followed by retained tissues and genital trauma. Our findings concur with previous studies highlighting that the main cause of primary PPH and the primary direct cause of maternal morbidity globally was uterine atony [6, 7, 18]. However, one other study concluded that genital tract laceration was the commonest cause of primary PPH followed by uterine atony [13]. The active management of the third stage of labour (AMSTL)with uterotonics was found to reduce the risk of PPH especially due to atonic uterus, and injectable oxytocin is the treatment recommended by WHO [37, 91].

Rwanda is applying WHO guidelines manage the third stage of labour. For all births (both vaginal and caesarean section births), it is recommended to use uterotonics to prevent postpartum haemorrhage (PPH) [91]. In spite of the widespread availability of oxytocin, some women included in our study were not given a uterotonic medication within three minutes of giving birth as an important component of AMTSL recommended by WHO [89] to prevent PPH. Our study demonstrated that 84.1% women among those who developed PPH had received an intramuscular uterotonic to manage the third stage of labour versus 90% in the control group. This rate is low compared to a prospective cohort study conducted in Uganda to understand the relative contributions of different risk factors for PPH. The results revealed that almost all (97%) women delivering at the health facilities health in rural Uganda received uterotonics. One of the factors affecting AMTSL practice identified in Tanzania, was incorrect time to administer oxytocin [92]. This is a context similar to Rwanda, where staff cannot have enough time to follow the procedure in an environment with limited staff performing multiple tasks [43]. This may be one of the reasons preventing certain women from receiving adequate uterotonic medication. This difference might be also associated with inadequate management of the third stage of labour as revealed by previous studies [4, 43, 93] in relation to factors affecting the prevention of PPH. This is also confirmed by findings from an endline evaluation of the 50,000 Happy Birthdays Project implemented in Rwanda between 2018–2020. The programme involved training midwives, nurses and other health workers to apply the Helping Mothers Survive (HMS) and Helping Babies Survive (HBS) techniques to address the leading causes of maternal and neonatal mortality including PPH. The baseline assessment demonstrated that 87.2% of women giving birth received uterotonics immediately after birth while after training, the endline evaluation revealed that 99.9% of women giving birth had received uterotonics for PPH prevention [94]. Beside medical interventions preventing PPH by usage of uterotonics, the experience of HMS simulation based training in Rwanda and in other similar settings also involve mechanical interventions such as bimanual uterine compression, the use of uterine balloon tamponade to manage and prevent PPH and its complications. These practices are commendable as they demonstrate successful knowledge and skill acquisition among frontline healthcare workers attending births, and improved clinical outcomes of childbearing women [94–96].

## Strengths and limitations

The characteristics and quality of the data source are the key aspects that give strength to this case-control study. The analysis of medical records in maternity units combined with structured interviews with participants during the hospital stay allowed to collect accurate data and minimized missing data. Case control studies are useful to study multiple exposures in the same outcome [40]. Hence, we were able to assess possible risk factors for primary PPH using a wide range of demographic and clinical data that were difficult to obtain from medical records. We were also able to record accurate information on the causes of PPH by reviewing medical records. It is important to note that the assessment of risk factors retrospectively is a shortcoming of this study. We chose as much as possible available cases with primary PPH and a random sample of controls from the same source population to reduce selection bias. Blood loss was estimated visually by health care providers in the five health facilities included in our study, and the blood loss may have been under or overvalued and this might have led also to missing some PPH cases [10]. To minimize this risk, we trained research assistants to identify cases of PPH based on clinical features and visual estimation of blood loss as recommended by Hancock *et al.*, [40] that the diagnosis and early detection of PPH may rely on factors other than volume.

## Conclusions

Primary PPH is a common occurrence in the Northern Province of Rwanda. Antepartum haemorrhage, multiple pregnancy and Hb Level <11 gr/dL on admission to labour were prevalent antepartum risk factors in primary PPH. During the intrapartum and immediate postpartum period, the main prevalent causes of PPH were uterine atony, retained tissues; and genital organ lacerations identified after birth. Since primary PPH is so prevalent, health care providers of the obstetric care units should be equipped to take care of clients who experience this complication. In Rwanda, progress toward improving maternal health would require concerted efforts to improve risk identification and promote adequate documentation of maternal information during the course of childbirth for an early identification of women at risk and prevention of PPH. Improving the quality of proactive prevention should be a priority for policy makers, health managers and service providers to reduce the high risks of maternal mortality and morbidity associated with PPH. Large scale studies are needed to investigate further potential PPH risk factors.

## Supporting information

**S1 Dataset.**
(DTA)

**S1 Checklist. STROBE checklist.**
(PDF)

## Acknowledgments

We acknowledge the University of Rwanda and Western University through Training Support Access Model Project for supporting this research. We would like to thank the administration of Byumba, Nemba, Ruli, Rutongo district hospitals and the in charge of Rutare medicalized health centre for granting us permission to conduct this study. We also acknowledge the research assistants for their support in data collection. Our sincere gratitude goes to Dr. Andrea Nove, technical director at Novametrics in UK for her professional guidance and English editing of the present paper.

## Author Contributions

**Conceptualization:** Oliva Bazirete, Manassé Nzayirambaho, Aline Umubyeyi, Marilyn Evans.

**Data curation:** Oliva Bazirete, Manassé Nzayirambaho, Innocent Karangwa, Marilyn Evans.

**Formal analysis:** Manassé Nzayirambaho, Innocent Karangwa, Marilyn Evans.

**Investigation:** Oliva Bazirete, Manassé Nzayirambaho.

**Methodology:** Oliva Bazirete, Manassé Nzayirambaho, Aline Umubyeyi, Innocent Karangwa, Marilyn Evans.

**Project administration:** Oliva Bazirete.

**Supervision:** Manassé Nzayirambaho, Marilyn Evans.

**Visualization:** Oliva Bazirete, Manassé Nzayirambaho, Innocent Karangwa, Marilyn Evans.

**Writing – original draft:** Oliva Bazirete, Manassé Nzayirambaho, Aline Umubyeyi, Innocent Karangwa, Marilyn Evans.

**Writing – review & editing:** Oliva Bazirete, Manassé Nzayirambaho, Innocent Karangwa, Marilyn Evans.

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
