## [Decision Letter · Decision Letter 0]

24 Nov 2021

PONE-D-21-17992Investigating and modelling risk factors for primary postpartum haemorrhage among childbearing women in the Northern Province of Rwanda: a case control studyPLOS ONE

Dear Dr. Bazirete

Thank you for submitting your manuscript to PLOS ONE. After careful consideration, we feel that it has merit but does not fully meet PLOS ONE’s publication criteria as it currently stands. Therefore, we invite you to submit a revised version of the manuscript that addresses the points raised during the review process.

We look forward to receiving your revised manuscript.

Kind regards,

Jianhong Zhou

Associate Editor

PLOS ONE

Journal Requirements:

2. PLOS requires an ORCID iD for the corresponding author in Editorial Manager on papers submitted after December 6th, 2016. Please ensure that you have an ORCID iD and that it is validated in Editorial Manager. To do this, go to ‘Update my Information’ (in the upper left-hand corner of the main menu), and click on the Fetch/Validate link next to the ORCID field. This will take you to the ORCID site and allow you to create a new iD or authenticate a pre-existing iD in Editorial Manager. Please see the following video for instructions on linking an ORCID iD to your Editorial Manager account: https://www.youtube.com/watch?v=_xcclfuvtxQ.

6. We noticed you have some minor occurrence of overlapping text with the following previous publication(s), which needs to be addressed:

- https://bmcpregnancychildbirth.biomedcentral.com/articles/10.1186/s12884-016-1217-0

The text that needs to be addressed involves the Strengths & limitations section.

In your revision ensure you cite all your sources (including your own works), and quote or rephrase any duplicated text outside the methods section. Further consideration is dependent on these concerns being addressed.

Reviewers' comments:

Reviewer's Responses to Questions

**Comments to the Author**

1. Is the manuscript technically sound, and do the data support the conclusions?

Reviewer #1: Yes

Reviewer #2: Yes

2. Has the statistical analysis been performed appropriately and rigorously? 

Reviewer #1: Yes

Reviewer #2: Yes

3. Have the authors made all data underlying the findings in their manuscript fully available?

Reviewer #1: Yes

Reviewer #2: Yes

4. Is the manuscript presented in an intelligible fashion and written in standard English?

Reviewer #1: Yes

Reviewer #2: Yes

5. Review Comments to the Author

Reviewer #1: This is a very important topic on PPH one of the most important causes of global maternal mortality.

Could you kindly review your manuscript taking the following into consideration

Under Methods, at the end of the second last paragraph clearly state that a P value of 0.05 was taken as statistical significant

Then round off all p values to 2 decimal places except those which are 0.0001. I note that you state that p values 0.077 are statistically significant, which is not correct. Rounded to 2 decimal places this is p=0.08 which is greater than p=0.05. Kindly correct this throughout your texts and report some factors that you reported as statistically significant when they are not.

Then also report your RR in 2 decimal places as well, neatly reporting them as e.g. (RR 6.7, 95% CI 4.48-9.38, p<0.001).

Reviewer #2: Herewith the authors of “Investigation and modelling risk factors for primary postpartum haemorrhage among childbearing women in the Northern Province of Rwanda: a case control study” are complemented about the investigation they performed.

Minor comments:

Title

The title is rather lengthy, reduction is suggested for example into “Risk factors for postpartum haemorrhage in Northern Province of Rwanda: a case control study”

Abstract, clear and concise, no comments

Introduction

Typo page 3, 2nd paragraph line 6: helps has to be help

Typo page 3, 2nd paragraph last sentence, “in Come countries”, has to be income countries.

Question: the authors are invited to elaborate on their adaptation of the PPH definition for the low and middle income countries. Not only in relation to the WHO definitions but also in relation to WHO near miss definitions. Motivate why they rightfully did not include the number of blood transfusions for PPH in their setting.

Methods

The authors are complimented on the stepwise set-up of the various studies (review, qualitative descriptive study, study to develop validated risk assessment tool) to optimize evaluation for this specific area in Rwanda; the direct data assessment by dedicated research assistant combined data from maternity records and structured interviews; attention to language barriers by translation to and fro from English, French and Kinyarwanda.

Question: the authors are invited to inform whether blood transfusions were scored within all the variables and also if the number of blood transfusions were scored. In the present description of the methods this was not clear, whereas in presenting the results “received blood transfusions” was seen in the Tables 1. If possible present the number of blood transfusions given per person (median, range). And reflect on the results in the discussion.

Discussion

The discussion has been written eloquently.

A few questions/ advices are presented below:

The Authors plea for more vigilance on the part of practitioners attending labour and birth to identify women at risk, to have adequate preparation and plan for early intervention to prevent PPH.

Question: the authors are invited to elaborate on this subject, by presenting own suggestions how to do this with their present knowledge, and/ or use of other investigators in similar settings, for example Fawcus 2019 presenting practical approached to managing PPH with limited resources.

Page 18-19 the authors advise regular antenatal check-ups to detect risk factors for PPH and refer to reference 82.

Question: Can the authors present their own advice on how regular these check-ups ideally should be, or should be strived for.

Page 20, the authors mention introduction of active management at the third stage of labour.

Question: the authors are in line with prior comments to add practical recommendations of their own or other investigators working in the same setting. Here is the possibility not only to medical intervention with AMSTL with oxytocin but also possible mechanical intervention with bimanual uterine compression/ balloon/condom tamponade. Use the opportunity in the discussion to evaluate the obtained knowledge of the use of training and retraining programs of Helping Mothers Survive and which items you/ literature (various publications on this subject including your reference 91 concerning data from Rwanda) suggest to be included in these training programs.

6. PLOS authors have the option to publish the peer review history of their article (what does this mean?). If published, this will include your full peer review and any attached files.

Reviewer #1: No

Reviewer #2: **Yes: **Prof. J.I.P. de Vries, M.D., Ph.D., gynaecologist-perinatologist

---

## [Author Response · Author response to Decision Letter 0]

2 Jan 2022

Dear editor,

Dear reviewers,

Happy new year, 

Thank you for sending the constructive reviews of our manuscript – we found them extremely helpful and have amended the manuscript to reflect the recommendations where possible. Below we indicate how we have responded to two reviewers’ comments. We trust that these amendments have improved the manuscript and look forward to hearing from you regarding acceptability for publication. 

Kindly note that the co author Innocent Karangwa changed his address. The new one is: 

University of Cape Town, Rondebosch, Cape Town, 7700, South Africa

Email address: innocent.karangwa@uct.ac.za

Yours Sincerely,

---

## [Decision Letter · Decision Letter 1]

26 Jan 2022

Risk factors for postpartum haemorrhage in the Northern Province of Rwanda: a case control study

PONE-D-21-17992R1

Dear Dr. Bazirete,

We’re pleased to inform you that your manuscript has been judged scientifically suitable for publication and will be formally accepted for publication once it meets all outstanding technical requirements.

Kind regards,

Nnabuike Chibuoke Ngene, Dip HIV Med; MMed(FamMed); FCOG; MMed(O&G); Ph.D

Academic Editor

PLOS ONE

Additional Editor Comments (optional):

Reviewers' comments:

Reviewer's Responses to Questions

**Comments to the Author**

1. If the authors have adequately addressed your comments raised in a previous round of review and you feel that this manuscript is now acceptable for publication, you may indicate that here to bypass the “Comments to the Author” section, enter your conflict of interest statement in the “Confidential to Editor” section, and submit your "Accept" recommendation.

Reviewer #1: All comments have been addressed

2. Is the manuscript technically sound, and do the data support the conclusions?

Reviewer #1: Yes

3. Has the statistical analysis been performed appropriately and rigorously? 

Reviewer #1: Yes

4. Have the authors made all data underlying the findings in their manuscript fully available?

Reviewer #1: Yes

5. Is the manuscript presented in an intelligible fashion and written in standard English?

Reviewer #1: Yes

6. Review Comments to the Author

Reviewer #1: The authors have improved the manuscript by using the suggestions, cleaning out the statistics, and using the correct decimal points.

7. PLOS authors have the option to publish the peer review history of their article (what does this mean?). If published, this will include your full peer review and any attached files.

Reviewer #1: **Yes: **Solwayo Ngwenya

---

## [Editor Report · Acceptance letter]

4 Feb 2022

PONE-D-21-17992R1 

Risk factors for postpartum haemorrhage in the Northern Province of Rwanda: a case control study 

Dear Dr. Bazirete:

I'm pleased to inform you that your manuscript has been deemed suitable for publication in PLOS ONE. Congratulations! Your manuscript is now with our production department. 

Kind regards, 

on behalf of

Dr. Nnabuike Chibuoke Ngene 

Academic Editor

PLOS ONE